# Hidden duplicates: 10s or 100s of Indian trials, registered with ClinicalTrials.gov, have not been registered in India, as required by law

**Sangeeta Kumari, Abhilash Mohan, Gayatri Saberwal**  *

Institute of Bioinformatics and Applied Biotechnology, Biotech Park, Bengaluru, Karnataka, India

* gayatri@ibab.ac.in

## Abstract

### Background

This study's primary goal was based on the fact that since 15 June 2009 it has been mandatory to register regulatory trials running in India with Clinical Trials Registry–India (CTRI). Were all such trials, registered with ClinicalTrials.gov (CTG) after 2009, that included India as a location, also registered with CTRI? We first had to determine how to correctly identify a trial that was registered in both the registries, but that lacked the relevant secondary ID. Therefore the secondary goal of this study was to identify the best method to do this.

### Methods

We used a control set of 1013 trials that cross-referenced a record in the other registry. We used two algorithms to–in a blinded fashion–identify CTRI matches for the 1013 CTG records. 80% of the predictions were correct. Using the same methodology, we identified matches for the CTG trials without known CTRI matches. We then used a logistic regression model to predict which of these matches were correct.

### Results

(i) 3664 CTG records listed India as a location, but did not list any CTRI ID, and were not identified by any CTRI records either. (ii) The best single field to find a CTRI match for a CTG trial was the title field. (iii) Between 50 and 300 of 581 relevant CTG trials were not registered with CTRI.

### Conclusions

This is the first study to use hidden duplicates to determine that the law on trial registration has been broken (in India). Similar studies need to be done for trials run in other countries.

**Data Availability Statement:** All relevant data are within the manuscript and its Supporting Information files.

**Funding:** GS received internal institutional funds. These were partially from the Government of

Karnataka's Department of Information Technology, Biotechnology and Science & Technology (https://itbtst.karnataka.gov.in/english). There was no grant number. The funders had no role in study design, data collection and analysis, decision to publish, or preparation of the manuscript.

**Competing interests:** The authors have declared that no competing interests exist.

## Introduction

Clinical trial registries were set up to increase both the visibility of planned or ongoing trials and the accountability and transparency of the trial enterprise. Today, there are dozens of such registries [1], set up by companies, governments, non-profits and universities. The best known public repository is that of the United States of America (US), ClinicalTrials.gov (CTG). The International Clinical Trials Registry Platform (ICTRP), established by the World Health Organization (WHO), enables users to search 18 open-access registries, including CTG. We visited the websites of each of these registries [2–19] on 15 April 2020, and determined that they cumulatively hold 598,746 records, with CTG holding 335,963 (56%).

Such registries were primarily set up to increase enrollment in trials, and to prevent a bias in the medical literature due to the selective reporting of positive study outcomes. Nevertheless, their records have been put to other uses as well, such as (i) examining whether the clinical research enterprise mirrors the disease burden of a nation [20–22], (ii) tracking the development of new medical technologies [23]; and (iii) identifying ethically questionable studies [24,25]. The records have also been used to determine whether the law has been broken, as illustrated by the following cases: (i) In the US, it is mandatory that sponsors deposit the results of most trials in CTG within one year of trial completion [26]. The website http://fdaaa.trialstracker.net uses registry data to track those who have not done so, thereby providing the government with actionable information on those who are breaking the law [27]. (ii) Health advocacy groups identified certain trials, run by a multinational company in India, that allegedly contravened the law. This led to the government ordering an investigation [28]. (iii) A study demonstrated that there were insufficient numbers of registered trials concerning the long-term safety of medical devices. If post-approval studies are mandated but not carried out, it is a matter for regulatory action [29].

Along similar lines, this study used data in two registries to ascertain whether Indian law has been broken. It's primary goal was based on the fact that since 15 June 2009 it has been mandatory to register regulatory trials running in India with Clinical Trials Registry–India (CTRI). Since a given trial may have been registered in multiple registries, some of these studies may also have been registered with CTG. Have all such trials registered with CTG, and with India as a location, also been registered with CTRI? In order to answer this question, we had to determine how to correctly identify a study that had been registered in both the registries, but that lacked the relevant secondary ID, issued by the other registry. Therefore, the secondary goal of this study was to identify the best possible method to do this. We were able to partially achieve both the goals of this study.

## Methods

On 15 March 2019, CTRI contained 18,074 records (S1 Table). There were two cases of duplicate CTRI numbers which represented separate studies, as described in S1 Table. We took forward the 18,072 unique CTRI numbers. We web-scraped the CTRI database using an in-house python script (S1 File, described in S1 Data), and downloaded the information in the secondary ID field from all the trial records. From these, we identified records that listed an ID issued by CTG, that is, an NCT number. After deleting 10 erroneous records, we obtained 968 CTRI records with a unique NCT ID. On the same day we also examined the CTG database, and identified 102 records that listed a CTRI number in the secondary ID field. 57 records were common to the two searches, leaving 1013 unique trials that were registered in either database. We referred to these 1013 pairs of records as 'true pairs' of trials, that is those that the trialists identified as the same trial. The steps to reach this set of records are summarized in Table 1, with details in S1 Table.

**Table 1. A summary of the process to identify 1013 true pairs of CTG-CTRI records.**

|   | Step | Number of records |
|---|------|-------------------|
| 1 | CTRI records with an NCT ID in the secondary ID field | 968 |
| 2 | CTG records with a CTRI ID in the secondary ID field | 102 |
| 3 | Total number of either CTRI or CTG records that identified each other | 1070 |
| 4 | CTRI and CTG records that both identified each other | 57 |
| 5 | Unique CTG-CTRI pairs, identified either from the CTG or CTRI records | 1013 |

We then used the python script to scrape 13 fields of information from the 1013 CTRI records. The fields were as follows: (i) Public title of study, (ii) Scientific title of study, (iii) Recruitment status of trial (India), (iv) Phase of trial, (v) Total sample size, (vi) Primary outcome, (vii) Intervention/comparator agent, (viii) Health condition/problems studied, (ix) Countries of recruitment, (x) Source of monetary or material support + Primary sponsor (subsequently concatenated), (xi) Study design and (xii) Type of trial. From the Primary Sponsor field, we only took the name, not the address or type of sponsor (S1 Table).

Similarly, we downloaded information from the corresponding 11 fields from the partner CTG record, as follows: (i) Title (which mapped to both the 'Public title of study' and 'Scientific title of study' of the CTRI record), (ii) Recruitment status, (iii) Phase, (iv) Number enrolled, (v) Outcome measures, (vi) Interventions, (vii) Conditions, (viii) Locations, (ix) Sponsor/Collaborators, (x) Study design and (xi) Study type (S1 Table). We did not use the 'Funder type' field, since this had only four categories (NIH; Other U.S. Federal agency; Industry; and All others (individuals, universities, organizations)), and therefore did not effectively distinguish between trials.

## Handling of missing values

In 350 of the 1013 CTG-CTRI pairs, one of the records was missing data in one or more fields. 'Not applicable' was the only one which was considered a valid option, and was not considered to be missing data. Entries such as na, not available, n/a, nil, NIL, and NOT AVAILABLE were replaced by a blank (S2 Table).

## String-match methodology

We then used two different types of algorithms to quantitatively compare the entries in the corresponding fields in the true pairs of CTG-CTRI records.

(a) We matched each field of the CTG-CTRI pair, using the term frequency inverse document frequency algorithm Tf-Idf (hereafter tfidf). Sentences were tokenized, and this was followed by word embedding. This algorithm converts words into vectors, and then calculates the cosine similarity of the vectors. We used the Scikit-learn (0.20v) implementation of tfidf.

(b) We also performed text matching of each field of the CTG-CTRI pair using the fuzzywuzzy string match algorithm. This is based on the edit distance algorithm, which assesses the number of changes that are needed to make one string identical to another. We used the fuzzy string match, implemented in python module fuzz. The function fuzz.token_set_ratio (hereafter, fuzz) was used to generate a score for each pair of fields in the 1013 true pairs.

As mentioned, we web-scraped 13 fields (reduced to 12 after concatenation, as explained above) from the CTRI records, downloaded the corresponding 11 fields from the CTG records, and matched the corresponding fields.

## Model for true-pair identification

Fuzz and tfidf each produced a 1013 x 12 matrix for the true pairs (S3 Table). Next, a dataset of false pairs was generated using the 1013 records from CTG and CTRI, by random sampling. Fuzz and tfidf each produced a 1013 x 12 matrix for these false pairs as well (S3 Table). We created a merged dataset of the true and false pairs for fuzz, and then performed label binarization using standard methods, before building a logistic regression model.

## Identifying important features

We used the resulting regression coefficient values, of true or false status of the matches, to identify five pairs of fields (including cases of concatenation) as the best performing ones (S4 Table). These pairs of CTG and *CTRI* fields, were, respectively: (i) Title and *Public title of study* and *Scientific title of study* (concatenated), (ii) Conditions and *Health Condition*, (iii) Interventions and *Intervention*, (iv) Outcome Measures and *Primary Outcome*, and (v) Sponsor/Collaborators and *Source of Monetary or Material Support + Primary Sponsor* (concatenated). The five fields that were found to be significant were selected, and used to create a simpler logistic regression model. The model accuracy evaluation criteria are described in S1 Text.

## Identifying CTRI matches for the set of 1013 CTG records using the fuzz and tfidf algorithms

Although we already knew the correct CTRI matches for the 1013 CTG records, we wished to check whether our string-matching methods picked up the correct matches. On 20 June 2019 we web-scraped all 19,533 records in the CTRI database, and obtained the data for the five fields identified above as the most significant. We then used different combinations of fields to text match the 1013 CTG records against these 19,533 records with the fuzz algorithm. The correctly predicted matches are tabulated in S5 Table. We found that a combination of four fields (title, condition, intervention and sponsor) provided the highest percentage (83%) of correct matches. However the title alone had a high percentage (80%) of matches, which was very close to that provided by four fields. Using tfidf, the title field gave a similar high match (81%), as detailed in S6 Table. Therefore, we decided to use both fuzz and tfidf, together with the title field, for the next step, in which we identified putative matches for the CTG records with unknown CTRI partners.

## Unknown match study

On 5 July 2019 there were 3734 CTG records with India as a location, i.e., as a country of recruitment. We wished to subtract the 1013 CTG records whose CTRI matches had been previously identified. However it turned out that 187 of the 1013 records did not list India as a location. As such 826 records were deleted from 3734, leaving 2908 CTG records with India as a location and with an unknown CTRI match. This set comprised expanded access studies (7 records), observational studies (418), and interventional trials (2483). We took forward the 2483 interventional trials, which we bifurcated into those that were first posted after 2009 (1640 records) and those that were first posted up to, and including, 2009 (843).

Next, we identified the regulatory trials among the the set of 1640 records. For this, we excluded trials in Phases 1 and 4, and those for which Phase was labeled 'not applicable'. This

left those in Early Phase 1 (8 cases), Phase 1|2 (56), Phase 2 (102), Phase 2|3 (72) and Phase 3 (343), which totaled to 581 trials. The steps taken to arrive at this set of trials are listed in Table 2.

Using the title field with the fuzz and tfidf algorithms, we sought matches for the 581 CTG records in the CTRI database of 19,533 records. These were further queried using the logistic regression model to determine whether they were true matches or not. The results are in file S7 Table.

Finally, we did two types of manual assessments of the matches that were predicted to be either correct or incorrect: (a) We took a small, random, sample of 25 pairs each from the pairs that fuzz and tfidf had identified, and from the common set that both had identified, and that the model had predicted were correct. For these 75 pairs, we did a manual assessment of six fields (Type of study, Sponsor, Condition, Intervention, Sites, and Phase) in both the CTG and CTRI records, to determine whether or not they appeared to be correct matches (S8 Table). (b) We followed the same process for 25 pairs each, that fuzz and tfidf had identified, and that the model had predicted were incorrect, to determine whether or not they appeared to be incorrect matches (S9 Table).

To be noted, all the codes written by SK were cross checked by AM. The results generated by the codes were also cross checked by AM. The manual crosschecking of all pairs was done by both SK and GS, independently.

## Results

By examining trials registered with CTRI that listed an NCT ID as secondary ID, and vice versa, we identified 1013 unique pairs of IDs that represented the same study. These 1013 trials were the 'positive controls' on which we built this study.

We went on to use 581 CTG trials to find matches in the Indian database, and used a model to predict whether the matches were correct. Using fuzz (or tfidf) and the title field, 581 CTG records picked up 581 matches. The model predicted 434 (75%) (or 441, 76%) of these to be correct. Since, in the control set, fuzz (tfidf) had predicted 80% (81%) of the correct matches, we extrapolated the figure 434 (441) to 543 (544) correct matches, leaving 38 (37) incorrect matches. However, of the 434 and 441 matches picked up by fuzz and tfidf, respectively, that were predicted to be correct, 288 were common. That left 293 incorrect matches. These results are summarized in TableS 3 and detailed in S8.

Finally, in the manual assessment, we first took a small, random, sample of 25 pairs each from the pairs that (i) fuzz and (ii) tfidf had identified, and from (iii) the common set that both had identified; and that the model predicted were correct. For these 75 pairs, we manually examined six fields in both the records, to ascertain whether or not they were truly correct (S9 Table). 48%, 72% and 68% of these matches, respectively, appeared to be correct. Next, we took a small, random, sample of 25 pairs each from the pairs that (i) fuzz and (ii) tfidf had identified, that the model predicted were incorrect. 100% and 88% of these matches, respectively, appeared to be incorrect.

**Table 2. Identifying the 581 CTG records with unknown CTRI matches.**

| | The nature of the records | Number of records |
|---|---|---|
| 1. | CTG records with India as a location, and with unknown CTRI matches | 2908 |
| 2. | Interventional trials | 2483 |
| 3. | Trials first posted after 2009 | 1640 |
| 4. | Regulatory trials, that is those in Early Phase 1, Phase 1|2, Phase 2, Phase 2|3 or Phase 3. | 581 |

**Table 3. Identifying CTRI matches for the 581 CTG records with unknown CTRI matches.**

| | The nature of the records | Number of records | Percentage of 581 |
|---|---|---|---|
| 1. | Using fuzz and the title field, 581 CTRI matches were picked up for the 581 CTG records. The model predicted 434 of these to be true. | 434 | 74.7% |
| 2. | Assuming that fuzz picked up 80% of the true matches, the number of correct matches would be 543. | 543 | 93.5% |
| 3. | Incorrect matches, as predicted by fuzz. | 38 | 6.5% |
| 4. | Using tfidf and the title field, 581 CTRI matches were picked up for the 581 CTG records. The model predicted 441 of these to be true. | 441 | 75.9% |
| 5. | Assuming that tfidf picked up 81% of the true matches, the number of correct matches would be 544. | 544 | 93.6% |
| 6. | Incorrect matches, as predicted by tfidf. | 37 | 6.4% |
| 7. | CTRI matches picked up by both fuzz and tfidf. | 304 | 52.3% |
| 8. | Of the 304 CTRI matches picked up by both fuzz and tfidf, the model predicted 288 to be true. | 288 | 49.6% |
| 9. | Incorrect matches from the common cases. | 293 | 50.4% |

## Discussion

In this study we wished to determine whether all regulatory trials running in India since 15 June 2009, and registered in the US, were also registered in India, as required by law. For this, we used known cases of studies that were registered in both the databases to identify the best methodology to correctly identify a CTRI trial that had also been registered with CTG but lacked the relevant CTRI ID in the latter record. We found that using the fuzz and tfidf algorithms, the title field was the best single field to identify the CTRI duplicate.

We discuss the phenomenon of duplicate registrations more generally before coming to the duplicates of this study. We go on to discuss the issue of trials required to be registered in India. Finally, we discuss the CTRI matches, and lack thereof, for CTG trials that did not list CTRI IDs in the secondary ID field.

### The issue of duplicate registrations

From the early years of the existence of registries, the issue of the duplicate registration of a given trial has received attention [30]. The meta-analyses of trial results, that lead to clinical guidelines, must be based on single records of each study. Otherwise, there will be a bias in favor of the results from records that are present in multiple registries. Aside from studies that may contribute to clinical guidelines, other analyses of large sets of registry records have also aimed to avoid double counting [22,31–33].

Duplicates identified by the listing of a relevant secondary ID have been referred to as 'known duplicates', and the others as 'hidden duplicates' [34]. Hidden duplicates are a particularly challenging issue. To prevent such duplicates, it has been suggested that each trial should receive a single ID, the Universal Trial Registration Number. However this has not happened. The idea of a single world-wide registry [35] has also not taken off. The ICTRP does direct its primary registries to (i) establish processes to prevent duplicates within a given database; and (ii) enable one or more secondary IDs to be entered in a study record, even retrospectively. It also makes recommendations on when a trial should or should not be registered in more than one registry [36]. These guidelines have recently been reiterated [37].

Since mid-June 2009, regulatory trials running in India must be registered with CTRI. Therefore, if such a study was registered in the US, but not in India, then Indian law had been broken. Usually one wishes for a situation of fewer hidden duplicates. However in this study we

hoped to find many of them, since the more hidden duplicates we found, the less the law had been broken. Further, if a study was indeed registered in both the registries, but neither of the records cross-referenced the other, then this fell short of the best practices of trial registration.

We have identified 2908 CTG records with India as a location and with an unknown CTRI match. The managers of CTRI could coordinate with the Indian drug regulator to pressure the relevant Responsible Parties to update their records with the Indian IDs, should the trials actually have them.

### India's rule since mid-2009

As mentioned above, since 15 June 2009 it has been mandatory to register regulatory trials running in India [38]. Should the responsible individual or organization fail to do so, the licensing authority is empowered to take actions such as issuing a warning letter, rejecting the results of the trial, or barring the investigator or sponsor from future trials for any length of time [39].

As such, the date of registration of a trial is important. For a given CTRI record, the date the study was registered was available. For a CTG record, the date of registration was unavailable, but the date the trial was first submitted to the registry was available. We also knew the date it was 'First posted', that is when it was first made available on the registry website, which could have been a few days or a few weeks after it was submitted. A trial registered with both the registries may have been registered in India either before or after the 'First submitted' or 'First posted' dates in CTG. In identifying US records of studies that ran in India after mid-June 2009, and which should have been registered with CTRI, we considered the fact that registration in one registry may be delayed. Accordingly, we provided two types of margins. We looked for (i) CTG trials based on their 'First posted' date rather than their 'First submitted' date, and (ii) trials first posted after 2009. We could have provided an even bigger margin, and considered trials posted after 2010 instead. However we believe that our conclusions would not have been significantly different since only 58 studies, which comprised just 10% of the 581 trials considered, were first posted during 2010.

### Identifying the subset of regulatory trials

The Government of India has defined a new drug, as follows [39]: (i) one that has not been used much in the country thus far, and has not been previously approved; (ii) an earlier approved drug for which there is a new dosage, dosage form, route of administration or indication; (iii) a novel fixed-dose combination, the components of which have already been approved individually; and (iv) any vaccine or any drug manufactured by recombinant DNA technology.

In order to identify trials for new drugs, we excluded those in phase 1, which may have included bioavailability and bioequivalence studies, and those in phase 4, which have been run after regulatory approval. We also excluded those for which phase was labeled 'not applicable'. This left those in Early Phase 1, Phase 1|2, Phase 2, Phase 2|3 and Phase 3, which we analyzed. BA/BE studies were unlikely to be called Early Phase 1 studies, and therefore we included the latter group of eight studies. Regardless of whether or not applications related to these candidate drugs were later submitted to India's Central Drugs Standard Control Organisation for approval, the trials were required to be registered in India [39].

### Estimating the number of CTG trials without CTRI matches

In looking for Indian matches of US records, we used two algorithms, fuzz and tfidf. We found that the title field was the best single field to identify matches. Others have also found the title to be the best field to identify hidden duplicates [34].

Fuzz (tfidf) identified 434 (441) matches that the model predicted were true. However, with the control set of 1013 records, we had demonstrated that using this algorithm only predicted 80% (81%) of the correct matches. Extrapolating from these figures, we expected fuzz to identify 543 (544) correct matches, leaving 38 (37) CTG records with incorrect CTRI matches. However, of the large number predicted to be correct, based on a small sample of manually checked records, a maximum of 72% were actually so. This would raise the number of incorrect matches from 38 to about 53. Another way of trying to assess how many matches were incorrect is as follows: Of the matches identified by fuzz and tfidf that were predicted to be correct, only 304 were common, of which 288 were predicted to be correct. Of the remaining 293, a small sample of manually checked records were largely demonstrated to be incorrect. Therefore, this study predicts that between 50 and 300 regulatory trials that ran in India after 2009, and were registered in the US, were not registered in India. Although 50–300 trials is a wide range, more important than the actual numbers is the fact of missing matches. The Indian regulator needs to launch an investigation, based on CTG records with India as a location, to determine whether particular sponsors have broken Indian law, and if so, to take necessary action.

Finally, this study had certain limitations, as follows: (i) The model was built using information in certain fields of each member of a CTG-CTRI pair. Since we had no means to determine the veracity of the inputted data, the model may have been built on somewhat imperfect data. (ii) We may have erred in including certain trials in the test sample of 581 that were initiated in India before June 2009 and were therefore not required to be registered, but there was no way to identify such studies. (iii) It is possible that some of the trials that we excluded, such as some of those in Phase 1, were regulatory trials, and therefore were required to be registered with CTRI. As such, the pool of studies that we investigated may have been somewhat smaller than warranted. (iv) It is possible that when a study was registered in the US, it was initially planned to be run in India, but subsequently there was a change in the international locations. The Responsible Party may not have updated the CTG record to reflect this. However, we had no way to determine whether this was so for any trial. (v) Since CTG is the largest of such public registries, we only looked for duplicates of Indian studies in this database. There may have been hidden duplicates in other registries, thereby expanding the universe of hidden duplicates of Indian trials.

## Conclusions

In conclusion, we have demonstrated that (i) there were 3664 US records that listed India as a location, but did not list a CTRI ID, and were not identified by any CTRI records either, (ii) using the title field, the fuzz and tfidf algorithms were good ways to find a true match of a US trial in CTRI; and (iii) between 50 and 300 trials, registered with CTG, were not registered with CTRI, although the law requires it. To the best of our knowledge, this is the first study to use hidden duplicates to establish that Indian law has been broken.

In general, it is important to minimize the number of hidden duplicates in order to (a) reduce the bias both in meta-analyses of trial results that will inform clinical guidelines, and in any other analysis of large sets of registry records, and (b) ascertain whether the relevant Indian (or any other) law has been broken. Going forward, registries should make it compulsory to provide the Universal Trial Number, as suggested by WHO's ISCTR [40]. Alternatively, and to reiterate the minimum standards recommended by ISCTR, they should require trialists to provide the relevant ID if the study is registered elsewhere. If it is not, ISCTR recommends that the secondary identifier field be filled with "Nil known". We suggest that this requirement be taken one step further. Trialists should explicitly confirm that the trial is not, and is unlikely

to be, registered elsewhere. Further, it should be possible for the registry to coordinate with the drug regulator to compel trialists to provide this information, even retrospectively. If all registries enforce such mechanisms to address this issue, then it will go a long way in reducing the number of hidden duplicates.

## Supporting information

**S1 File. All the scripts used to process the data of this study.**
(PY)

**S1 Data. Legends for all the scripts used to process the data.**
(DOC)

**S1 Text. Details of the logistic regression model coefficient and model accuracy.** An explanation of (a) what the coefficient of logistic regression implies, and (b) the model accuracy evaluation criteria.
(DOC)

**S1 Table. Identifying the 1013 unique CTRI records with valid NCT IDs.** The steps taken from all the CTRI records to the 1013 unique CTRI-CTG pairs, identified either from the CTRI or the CTG records.
(XLSX)

**S2 Table. The field-wise missing data in the CTG and CTRI records.** Both the field-wise missing data in the CTG and CTRI records, and the enumeration of the CTG and CTRI records where there was at least one missing value in a given field.
(XLS)

**S3 Table.** A quantitative comparison of the corresponding fields in the (a) true and (b) false CTG-CTRI pairs, using the tfidf and fuzz algorithms. Fuzz and tfidf each produced a matrix with dimension 1013 x 12 for the true pairs. A dataset of false pairs was generated using the 1013 records from CTG and CTRI, by random sampling. Fuzz and tfidf each produced a matrix with dimension 1013 x 12 for these false pairs as well.
(XLS)

**S4 Table. The regression coefficients of 12 variables obtained by the logistic regression model.** By using their regression coefficient values to identify the 0 or 1 status i.e whether they are true or false pairs, five pairs of fields (including two cases of concatenation of two fields each) out of 12, were identified as the best performing ones.
(XLS)

**S5 Table. Identifying CTRI matches for the set of 1013 CTG records using the algorithms.** Data from varying numbers of field(s) of the 1013 CTG records were used to pick up the best matches in the set of 19,533 CTRI records. The percentage of correct matches could be determined because we already knew the correct CTRI matches.
(XLS)

**S6 Table. Using the title field, and the fuzz or tfidf algorithm, to identify the correct matches for the 1013 CTG records in the database of 19,533 records.** Ignoring the fact that we already knew the correct matches, we used the 1013 CTG records to identify their best matches in the entire database of 19,533 records.
(XLSX)

**S7 Table. Using fuzz and tfidf to find true CTRI matches for CTG cases with India as a location but no known CTRI matches.** We took 581 'regulatory' CTG trials and looked for their best matches in the 19,533 records of the CTRI database.
(XLSX)

**S8 Table. A manual assessment of six fields from a sample of 75 CTG-CTRI pairs that the model predicted were correct.** We manually examined 25 pairs each from the sets that fuzz and tfidf identified, and 25 from the set that both algorithms identified, to determine whether or not the pairs appeared to be truly correct.
(XLS)

**S9 Table. The list of the 293 incorrect NCT-CTRI matches, and a manual assessment of six fields from a sample of 50 CTG-CTRI pairs that the model predicted were incorrect.** We manually examined 25 pairs each from the sets that fuzz and tfidf identified, to determine whether or not the pairs appeared to be truly incorrect.
(XLS)

## Acknowledgments

We are grateful to Dr. S. Thiyagarajan and Dr. R. Srivatsan, from IBAB, for discussions.

## Author Contributions

**Conceptualization:** Gayatri Saberwal.

**Data curation:** Sangeeta Kumari.

**Formal analysis:** Sangeeta Kumari, Abhilash Mohan.

**Funding acquisition:** Gayatri Saberwal.

**Investigation:** Sangeeta Kumari.

**Methodology:** Sangeeta Kumari.

**Project administration:** Gayatri Saberwal.

**Resources:** Gayatri Saberwal.

**Software:** Sangeeta Kumari.

**Supervision:** Gayatri Saberwal.

**Validation:** Abhilash Mohan, Gayatri Saberwal.

**Visualization:** Sangeeta Kumari.

**Writing – original draft:** Gayatri Saberwal.

**Writing – review & editing:** Sangeeta Kumari, Abhilash Mohan, Gayatri Saberwal.

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
