## [Decision Letter · Decision Letter 0]

10 Apr 2020

PONE-D-20-02492

Hidden duplicates: At least 3 dozen Indian trials, registered with ClinicalTrials.gov, have not been registered in India, as required by law

PLOS ONE

Dear Dr. Saberwal,

Thank you for submitting your manuscript to PLOS ONE. After careful consideration, we feel that it has merit but does not fully meet PLOS ONE’s publication criteria as it currently stands. Therefore, we invite you to submit a revised version of the manuscript that addresses the points raised during the review process.

We would appreciate receiving your revised manuscript by May 24 2020 11:59PM. To enhance the reproducibility of your results, we recommend that if applicable you deposit your laboratory protocols in protocols.io, where a protocol can be assigned its own identifier (DOI) such that it can be cited independently in the future. For instructions see: http://journals.plos.org/plosone/s/submission-guidelines#loc-laboratory-protocols

We look forward to receiving your revised manuscript.

Kind regards,

Omid Beiki, M.D., Ph.D.

Academic Editor

PLOS ONE

Journal Requirements:

Reviewers' comments:

Reviewer's Responses to Questions

**Comments to the Author**

1. Is the manuscript technically sound, and do the data support the conclusions?

Reviewer #1: Yes

Reviewer #2: Partly

2. Has the statistical analysis been performed appropriately and rigorously? 

Reviewer #1: Yes

Reviewer #2: N/A

3. Have the authors made all data underlying the findings in their manuscript fully available?

Reviewer #1: Yes

Reviewer #2: Yes

4. Is the manuscript presented in an intelligible fashion and written in standard English?

Reviewer #1: Yes

Reviewer #2: No

5. Review Comments to the Author

Reviewer #1: The aim of this study was to investigate whether all such trials registered with CTG, that included India as a location, also been registered with CTRI or not. The author reported three dozen of the 581 relevant CTG trials were not registered with CTRI.

The strength of this study was to investigate the theme extensively. However, there were some weakness in this study.

First, the authors had better to add references to the following sentences in Introduction: “The biggest public repository is that of the United States of America (US), ClinicalTrials.gov (CTG), which holds over 320,000 records. The International Clinical Trials Registry Platform (ICTRP), established by the World Health Organization (WHO), enables users to search 18 open-access registries, including CTG. These registries cumulatively hold over 550,000 records.”

Second, they had better to add references to the following recommendations in Conclusions: “Going forward, registries should make it compulsory to provide the Universal Trial Number. Alternatively, they should require trialists to provide the relevant ID if the study is registered elsewhere, or to confirm that it is not, and is unlikely to be, registered elsewhere. Further, the registry could coordinate with the drug regulator to compel trialists to provide this information, even retrospectively. If all registries enforce this, then it will go a long way in reducing the number of hidden duplicates, which inter alia will make it easier to identify whether or not the law has been broken in this manner.”

Third, they had better to add description about penalties if the law about trial registration in India is broken.

Reviewer #2: The manuscript by Kumari et al. “Hidden duplicates: At least 3 dozen Indian trials, registered with ClinicalTrials.gov, have not been registered in India, as required by law” uses an informatics approach to identify clinical trials with India as location that are registered with the US federal register ClinicalTrials.gov (CTG in the manuscript) but not the Indian Clinical Trials Registry (CTRI). The manuscript is written in a rather convoluted way and the significance of the findings is not quite evident.

Major concerns

1) The authors describe exclusion of what they call “Phase 1” from their analysis (Page 22, Line 316) but then they say they have included “Early Phase 1” (Page 22, Line 319). The difference is not clear and should be further explained.

2) The authors used 1) a training set of clinical trials for their algorithms and 2) several exclusion/inclusion criteria (e.g. studies posted to CTG after 2009) to come up with 581 studies registered in CTG with India as location and tried to determine the number so these trials that were also registered with CTRI. It is intriguing and worrisome that common hits from their two different algorithms were only 304 (52.3%). Although they claim “at least 3 dozen Indian trials” do not conform with the law, this number is not at all informative. Did they try to manually identify the 293 cases that are not common hits of their algorithms? They keep repeating in the discussion that 37-38 trials predicted to be registered only in CTG is a significant underestimate but no effort has been made to come up with a more precise number.

Minor concerns

1) Page 12, Lines 96-98: The sentence “57 records were common to the two searches, leaving 1013 unique trials that were registered in both databases (S1 Table)” should read “57 records were common to the two searches, leaving 1013 unique trials that were registered in either database (S1 Table)”.

2) Page 13, Line 114: Status should read “Recruitment status”

3) Several of the subheadings in the Results and Discussion sections are not informative (e.g. “The Year 2009”) and should be re-written.

3) The bibliography is sloppily written and I would suggest the authors make sure it is presented in a uniform style. Additionally, I would suggest the following substitutions/changes/additions.

Page 10, Line 63: Add reference (Zarin DA, Tse T, Williams RJ, Carr S. Trial Reporting in ClinicalTrials.gov - The Final Rule. N Engl J Med. 2016;375(20):1998-2004. doi: 10.1056/NEJMsr1611785. PubMed PMID: WOS:000387856100015) after the sentence “In the US, it is mandatory that the results of most trials be deposited in CTG within one year of completion of the trial.”

Reference 6: Substitute reference (Turner L. ClinicalTrials.gov, stem cells and 'pay-to-participate' clinical studies. Regen Med. 2017;12(6):705-19. doi: 10.2217/rme-2017-0015. PubMed PMID: WOS:000417040900014) for the “Washington Post” article.

Reference 7: This is a book by the National Academies Press (https://www.nap.edu/catalog/12900/transforming-clinical-research-in-the-united-states-challenges-and-opportunities), cite appropriately.

Reference 10: Add complete reference.

6. PLOS authors have the option to publish the peer review history of their article (what does this mean?). If published, this will include your full peer review and any attached files.

Reviewer #1: Yes: Masahiro Banno

Reviewer #2: No

---

## [Author Response · Author response to Decision Letter 0]

28 Apr 2020

5. Review Comments to the Author

Reviewer #1: The aim of this study was to investigate whether all such trials registered with CTG, that included India as a location, also been registered with CTRI or not. The author reported three dozen of the 581 relevant CTG trials were not registered with CTRI.

The strength of this study was to investigate the theme extensively. However, there were some weakness in this study.

- - - - - - - - - - - - - - - - - - - - - - - - - - - - - - - - - - - - - - - - - - 

First, the authors had better to add references to the following sentences in Introduction: “The biggest public repository is that of the United States of America (US), ClinicalTrials.gov (CTG), which holds over 320,000 records. The International Clinical Trials Registry Platform (ICTRP), established by the World Health Organization (WHO), enables users to search 18 open-access registries, including CTG. These registries cumulatively hold over 550,000 records.”

Authors’ response: There were no references for these statements. We ascertained the numbers by visiting each website. We have now clarified this in the text, as follows:

“The best known public repository is that of the United States of America (US), ClinicalTrials.gov (CTG). The International Clinical Trials Registry Platform (ICTRP), established by the World Health Organization (WHO), enables users to search 18 open-access registries, including CTG. We have visited the websites of each of these registries, and determined that they cumulatively hold 598,746 records, with CTG holding 335,963 (56%).”

- - - - - - - - - - - - - - - - - - - - - - - - - - - - - - - - - - - - - - - - - - - - - - - - - 

Second, they had better to add references to the following recommendations in Conclusions: “Going forward, registries should make it compulsory to provide the Universal Trial Number. Alternatively, they should require trialists to provide the relevant ID if the study is registered elsewhere, or to confirm that it is not, and is unlikely to be, registered elsewhere. Further, the registry could coordinate with the drug regulator to compel trialists to provide this information, even retrospectively. If all registries enforce this, then it will go a long way in reducing the number of hidden duplicates, which inter alia will make it easier to identify whether or not the law has been broken in this manner.”

Authors’ response: We have modified the para, as follows: 

“Going forward, registries should make it compulsory to provide the Universal Trial Number, as suggested by WHO’s ISCTR [International Standards for Clinical Trial Registries –Version 3.0. Geneva: World Health Organization; 2018. Licence: CCBY-NC-SA3.0IGO. Accessed 16 April 2020]. Alternatively, they should require trialists to provide the relevant ID if the study is registered elsewhere. If it is not, we suggest that trialists should explicitly confirm that the trial is not, and is unlikely to be, registered elsewhere. Further...”

- - - - - - - - - - - - - - - - - - - - - - - - - - - - - - - - - - - - - - - - - - - - - - - - -

Third, they had better to add description about penalties if the law about trial registration in India is broken.

Authors’ response: Based on the other reviewer’s comment to make sub-headings more informative, we have renamed the Discussion section ‘The Year 2009’ to ‘India’s rule since mid-2009’. In this section, we have added the following:

As mentioned above, since 15 June 2009 it has been mandatory to register regulatory trials running in India [Pandey et al. Challenges in Administering a Clinical Trials Registry:

Lessons from the Clinical Trials Registry-India; 2013. Pharm Med (2013) 27:83–93

DOI 10.1007/s40290-013-0009-3]. Should the responsible individual or organization fail to do so, the Licensing Authority is empowered to take actions such as issuing a warning letter, rejecting the results of the trial, or barring the investigator or sponsor from future trials for any length of time [Ministry of Health and Family Welfare. Govt of India. Notification. The Gazette of India: Extraordinary, Part II, Section 3, Subsection (i), New Delhi: 2019 Mar 19 [cited 2019 Apr 21], pgs 1-264. Available from: https://cdsco.gov.in/opencms/export/sites/CDSCO_WEB/Pdf-documents/NewDrugs_CTRules_2019.pdf].

- - - - - - - - - - - - - - - - - - - - - - - - - - - - - - - - - - - - - - - - - - - - - - - - -

Reviewer #2: The manuscript by Kumari et al. “Hidden duplicates: At least 3 dozen Indian trials, registered with ClinicalTrials.gov, have not been registered in India, as required by law” uses an informatics approach to identify clinical trials with India as location that are registered with the US federal register ClinicalTrials.gov (CTG in the manuscript) but not the Indian Clinical Trials Registry (CTRI). The manuscript is written in a rather convoluted way and the significance of the findings is not quite evident.

Authors’ response:

We have reworked the manuscript, and hope that it is less convoluted now. We have also expanded upon the significance, explained in another response below.

- - - - - - - - - - - - - - - - - - - - - - - - - - - - - - - - - - - - - - - - - - - - - - - - -

Major concerns

1) The authors describe exclusion of what they call “Phase 1” from their analysis (Page 22, Line 316) but then they say they have included “Early Phase 1” (Page 22, Line 319). The difference is not clear and should be further explained.

Authors’ response: In the original manuscript, we had stated “In order to identify trials for new drugs, we excluded those in phase 1, which may have included bioavailability and bioequivalence studies...”. We felt that BA/BE studies were unlikely to be called Early Phase 1 studies. In any case, there are only eight of such trials, out of 581, so their contribution to the results is minor.

We have added the line “BA/BE studies were unlikely to be called Early Phase 1 studies, and therefore we included the latter group of eight studies. ”.

- - - - - - - - - - - - - - - - - - - - - - - - - - - - - - - - - - - - - - - - - - - - - - - - -

2) The authors used 1) a training set of clinical trials for their algorithms and 2) several exclusion/inclusion criteria (e.g. studies posted to CTG after 2009) to come up with 581 studies registered in CTG with India as location and tried to determine the number so these trials that were also registered with CTRI. It is intriguing and worrisome that common hits from their two different algorithms were only 304 (52.3%). Although they claim “at least 3 dozen Indian trials” do not conform with the law, this number is not at all informative. Did they try to manually identify the 293 cases that are not common hits of their algorithms? They keep repeating in the discussion that 37-38 trials predicted to be registered only in CTG is a significant underestimate but no effort has been made to come up with a more precise number.

Authors’ response: We have now listed the 293 cases that are not common hits in Table 12. We have also manually checked a total of 50 pairs of them, to confirm that they are largely (47/50) incorrect matches. 

In the initial manuscript, we were conservative in sticking to the figure of 37–38 hidden duplicates. We have now provided the entire range that the data reveals, that is roughly 50–300. We have also changed the title of the manuscript to reflect this.

We have expanded upon the significance, as follows: “Although 50–300 trials is a wide range, more important than the actual numbers is the fact of missing matches. The Indian regulator needs to launch an investigation, based on CTG records with India as a location, to determine whether particular sponsors have broken Indian law, and if so, to take necessary action.”

- - - - - - - - - - - - - - - - - - - - - - - - - - - - - - - - - - - - - - - - - - - - - - - - -

Minor concerns

1) Page 12, Lines 96-98: The sentence “57 records were common to the two searches, leaving 1013 unique trials that were registered in both databases (S1 Table)” should read “57 records were common to the two searches, leaving 1013 unique trials that were registered in either database (S1 Table)”.

Authors’ response: We have changed it.

- - - - - - - - - - - - - - - - - - - - - - - - - - - - - - - - - - - - - - - - - - - - - - - - -

2) Page 13, Line 114: Status should read “Recruitment status”

Authors’ response: In the Brief View of a record, and in the downloaded data of a given trial, it is just called status. Nevertheless, the full form is more informative, and we have changed it.

- - - - - - - - - - - - - - - - - - - - - - - - - - - - - - - - - - - - - - - - - - - - - - - - -

3) Several of the subheadings in the Results and Discussion sections are not informative (e.g. “The Year 2009”) and should be re-written.

Authors’ response: We have changed some of the sub-headings.

- - - - - - - - - - - - - - - - - - - - - - - - - - - - - - - - - - - - - - - - - - - - - - - - -

3) The bibliography is sloppily written and I would suggest the authors make sure it is presented in a uniform style. Additionally, I would suggest the following substitutions/changes/additions.

Page 10, Line 63: Add reference (Zarin DA, Tse T, Williams RJ, Carr S. Trial Reporting in ClinicalTrials.gov - The Final Rule. N Engl J Med. 2016;375(20):1998-2004. doi: 10.1056/NEJMsr1611785. PubMed PMID: WOS:000387856100015) after the sentence “In the US, it is mandatory that the results of most trials be deposited in CTG within one year of completion of the trial.”

Reference 6: Substitute reference (Turner L. ClinicalTrials.gov, stem cells and 'pay-to-participate' clinical studies. Regen Med. 2017;12(6):705-19. doi: 10.2217/rme-2017-0015. PubMed PMID: WOS:000417040900014) for the “Washington Post” article.

Reference 7: This is a book by the National Academies Press (https://www.nap.edu/catalog/12900/transforming-clinical-research-in-the-united-states-challenges-and-opportunities), cite appropriately.

Reference 10: Add complete reference.

Authors’ response: We have made these, and other, improvements to the list of References.

- - - - - - - - - - - - - - - - - - - - - - - - - - - - - - - - - - - - - - - - - - - - - - - - -

---

## [Decision Letter · Decision Letter 1]

2 Jun 2020

PONE-D-20-02492R1

Hidden duplicates: 10s or 100s of Indian trials, registered with ClinicalTrials.gov, have not been registered in India, as required by law

PLOS ONE

Dear Dr. Saberwal,

Thank you for submitting your manuscript to PLOS ONE. After careful consideration, we feel that it has merit but does not fully meet PLOS ONE’s publication criteria as it currently stands. Therefore, we invite you to submit a revised version of the manuscript that addresses the points raised during the review process.

We look forward to receiving your revised manuscript.

Kind regards,

Omid Beiki, M.D., Ph.D.

Academic Editor

PLOS ONE

Reviewers' comments:

Reviewer's Responses to Questions

**Comments to the Author**

1. If the authors have adequately addressed your comments raised in a previous round of review and you feel that this manuscript is now acceptable for publication, you may indicate that here to bypass the “Comments to the Author” section, enter your conflict of interest statement in the “Confidential to Editor” section, and submit your "Accept" recommendation.

Reviewer #1: (No Response)

2. Is the manuscript technically sound, and do the data support the conclusions?

Reviewer #1: No

3. Has the statistical analysis been performed appropriately and rigorously? 

Reviewer #1: Yes

4. Have the authors made all data underlying the findings in their manuscript fully available?

Reviewer #1: Yes

5. Is the manuscript presented in an intelligible fashion and written in standard English?

Reviewer #1: Yes

6. Review Comments to the Author

Reviewer #1: The authors revised the manuscript extensively. However, I have some comments.

First, The authors had better to clarify when they visited the website in the following sentence of Introduction: “We have visited the websites of each of these registries, and determined that they cumulatively hold 598,746 records, with CTG holding 335,963 (56%).”

Second, they had better to add reference to the following sentence of Introduction: “We have visited the websites of each of these registries, and determined that they cumulatively hold 598,746 records, with CTG holding 335,963 (56%).” I consider the sentence need reference about the web site they visited. The example of reference is as follows: World Health Organization. International Clinical Trials Registry Platform (ICTRP): About the WHO ICTRP. 2019. Available at https://www.who.int/ictrp/about/en/. Accessed August 4, 2019.

Third, they had better to add justifications or references about the following each sentence (one by one) in Conclusions: “Alternatively, they should require trialists to provide the relevant ID if the study is registered elsewhere. If it is not, we suggest that trialists should explicitly confirm that the trial is not, and is unlikely to be, registered elsewhere. Further, the registry could coordinate with the drug regulator to compel trialists to provide this information, even retrospectively. If all registries enforce this, then it will go a long way in reducing the number of hidden duplicates, which, inter alia, will make it easier to identify whether or not the law has been broken in this manner.” They had better to add explanation or references to connect Discussion with these recommendations in Conclusions. I think these recommendations in Conclusions appeared abruptly.

7. PLOS authors have the option to publish the peer review history of their article (what does this mean?). If published, this will include your full peer review and any attached files.

Reviewer #1: Yes: Masahiro Banno, MD, PhD

---

## [Author Response · Author response to Decision Letter 1]

4 Jun 2020

6. Review Comments to the Author

Reviewer #1: 

First, The authors had better to clarify when they visited the website in the following sentence of Introduction: “We have visited the websites of each of these registries, and determined that they cumulatively hold 598,746 records, with CTG holding 335,963 (56%).”

Authors’ response: We visited all the websites on 15 April 2020, and have modified the text accordingly.

- - - - - - - - - - - - - - - - - - - - - - - - - - - - - - - -

Second, they had better to add reference to the following sentence of Introduction: “We have visited the websites of each of these registries, and determined that they cumulatively hold 598,746 records, with CTG holding 335,963 (56%).” I consider the sentence need reference about the web site they visited. The example of reference is as follows: World Health Organization. International Clinical Trials Registry Platform (ICTRP): About the WHO ICTRP. 2019. Available at https://www.who.int/ictrp/about/en/. Accessed August 4, 2019.

Authors’ response: We have added the references for all 18 registries. Since a registry is slightly different from ICTRP, we consulted a document entitled ‘How to cite a record on a clinical trials register’ (https://www.who.int/docs/default-source/documents/health-topics/how-to-cite.pdf?sfvrsn=d027a259_2) to determine the format for referencing the registries.

- - - - - - - - - - - - - - - - - - - - - - - - - - - - - - - -

Third, they had better to add justifications or references about the following each sentence (one by one) in Conclusions: “Alternatively, they should require trialists to provide the relevant ID if the study is registered elsewhere. If it is not, we suggest that trialists should explicitly confirm that the trial is not, and is unlikely to be, registered elsewhere. Further, the registry could coordinate with the drug regulator to compel trialists to provide this information, even retrospectively. If all registries enforce this, then it will go a long way in reducing the number of hidden duplicates, which, inter alia, will make it easier to identify whether or not the law has been broken in this manner.” They had better to add explanation or references to connect Discussion with these recommendations in Conclusions. I think these recommendations in Conclusions appeared abruptly.

Authors’ response: We agree that the Conclusions appeared abruptly, and thank the reviewer for this comment. We have redone this section of the Conclusions (provided below), and it reads better now. However we were not able to provide references for two sentences, since they are our own suggestions (which we indicate or imply). We hope that the reviewer is happy with the flow now.

Original paragraph of the Conclusions:

Going forward, registries should make it compulsory to provide the Universal Trial Number, as suggested by WHO’s ISCTR [22]. Alternatively, they should require trialists to provide the relevant ID if the study is registered elsewhere. If it is not, we suggest that trialists should explicitly confirm that the trial is not, and is unlikely to be, registered elsewhere. Further, the registry could coordinate with the drug regulator to compel trialists to provide this information, even retrospectively. If all registries enforce this, then it will go a long way in reducing the number of hidden duplicates, which, inter alia, will make it easier to identify whether or not the law has been broken in this manner.

Re-written para:

In general, it is important to minimize the number of hidden duplicates in order to (a) reduce the bias both in meta-analyses of trial results that will inform clinical guidelines, and in any other analysis of large sets of registry records, and (b) ascertain whether the relevant Indian (or any other) law has been broken. Going forward, registries should make it compulsory to provide the Universal Trial Number, as suggested by WHO’s ISCTR [40]. Alternatively, and to reiterate the minimum standards recommended by ISCTR, they should require trialists to provide the relevant ID if the study is registered elsewhere. If it is not, ISCTR recommends that the secondary identifier field be filled with “Nil known”. We suggest that this requirement be taken one step further. Trialists should explicitly confirm that the trial is not, and is unlikely to be, registered elsewhere. Further, it should be possible for the registry to coordinate with the drug regulator to compel trialists to provide this information, even retrospectively. If all registries enforce such mechanisms to address this issue, then it will go a long way in reducing the number of hidden duplicates.

- - - - - - - - - - - - - - - - - - - - - - - - - - - - - - - -

---

## [Editor Report · Decision Letter 2]

5 Jun 2020

Hidden duplicates: 10s or 100s of Indian trials, registered with ClinicalTrials.gov, have not been registered in India, as required by law

PONE-D-20-02492R2

Dear Dr. Saberwal,

We’re pleased to inform you that your manuscript has been judged scientifically suitable for publication and will be formally accepted for publication once it meets all outstanding technical requirements.

Kind regards,

Omid Beiki, M.D., Ph.D.

Academic Editor

PLOS ONE
---

## [Editor Report · Acceptance letter]

9 Jun 2020

PONE-D-20-02492R2 

Hidden duplicates: 10s or 100s of Indian trials, registered with ClinicalTrials.gov, have not been registered in India, as required by law 

Dear Dr. Saberwal:

I'm pleased to inform you that your manuscript has been deemed suitable for publication in PLOS ONE. Congratulations! Your manuscript is now with our production department. 

Kind regards, 

on behalf of

Dr. Omid Beiki 

Academic Editor

PLOS ONE